# Transcriptomic Analysis of *Neocaridina denticulata sinensis* Gills Following FPPS Knockdown Reveals Its Regulatory Role in Immune Response

**DOI:** 10.3390/ijms26010065

**Published:** 2024-12-25

**Authors:** Hongrui Li, Dandan Feng, Chunyu Zhang, Mengfei Liu, Zixuan Wu, Yuke Bu, Jiquan Zhang, Yuying Sun

**Affiliations:** 1School of Life Sciences, Hebei Basic Science Center for Biotic Interaction, Hebei University, Baoding 071002, China; hongruili2024@163.com (H.L.); feng_18245151652@163.com (D.F.); chunyuzhang0124@163.com (C.Z.); 13230612530@163.com (M.L.); zxwu0527@163.com (Z.W.); b18453701822@163.com (Y.B.); 2Institute of Life Science and Green Development, Hebei University, Baoding 071002, China

**Keywords:** *Neocaridina denticulata sinensis*, farnesyl pyrophosphate synthase, RNAi, transcriptome, immunity

## Abstract

Farnesyl pyrophosphate synthase (FPPS) is a key enzyme in the terpenoid biosynthesis pathway, responsible for converting isopentenyl pyrophosphate (IPP) and dimethylallyl pyrophosphate (DMAPP) into farnesyl pyrophosphate (FPP). In crustaceans, FPPS plays an important role in various physiological processes, particularly in synthesizing the crustacean-specific hormone methyl farnesoate (MF). This study analyzed the evolutionary differences in the physicochemical properties, subcellular localization, gene structure, and motif composition of FPPS in *Neocaridina denticulata sinensis* (named NdFPPS) compared to other species. The significant evolutionary divergence of FPPS was observed in crustaceans, likely linked to its role in MF synthesis. After the RNA interference (RNAi)-mediated knockdown of *NdFPPS*, transcriptomic analysis of gills revealed the significant enrichment of differentially expressed genes (DEGs) in pathways related to metabolism and immunity. Gene set enrichment analysis (GSEA) showed that most of these immune-related pathways were significantly suppressed, suggesting that *NdFPPS* may indirectly regulate the immune response by modulating metabolic levels. During the early stages of *Vibrio parahaemolyticus* infection, the expression of *NdFPPS* in the gills was significantly downregulated and subsequently returned to its original levels. Overall, our results provide new perspectives on the role of FPPS in immune regulation and enrich the functional information of FPPS.

## 1. Introduction

Farnesyl pyrophosphate synthase (FPPS) is a critical enzyme that catalyzes the reaction of isopentenyl pyrophosphate (IPP) and dimethylallyl pyrophosphate (DMAPP) to produce farnesyl pyrophosphate (FPP) [1]. FPP is an essential intermediate in the terpenoid biosynthetic pathway, serving as a precursor for various bioactive compounds, including sesquiterpenes, diterpenes, and methyl farnesoate (MF), a compound specific to crustaceans. MF is functionally similar to juvenile hormone III (JH-III) in insects, playing a key role in regulating processes such as molting, reproduction, and sexual maturation [2]. FPPS functions as the rate-limiting enzyme in the biosynthetic pathway of MF, and its activity directly determines the synthesis level of MF, thereby regulating molting cycles, sexual maturation, and reproductive system development in crustaceans [3,4]. In recent years, the expression patterns of *FPPS* in various tissues of crustaceans and its regulatory functions have received considerable attention. Studies have shown that the knockdown of *FPPS* significantly affects ovarian metabolism and neuropeptide expression, indicating a direct regulatory role of FPPS in the reproductive system of crustaceans [5].

Although the role of FPPS in reproduction and hormone regulation in crustaceans has been extensively studied, its function in other critical tissues, such as gills, remains unclear. Recent findings indicate that *FPPS* is highly expressed in the gills of *Neocaridina denticulata sinensis* [5]. As a crucial immune organ in crustaceans, the gills are rich in various immune cells, particularly phagocytic hemocytes. These hemocytes can recognize, engulf, and eliminate invading pathogens. In addition, other immune cells in the gills contribute to immune regulation by secreting cytokines and antimicrobial peptides, such as crustins and lysozymes [6,7]. The immune responses in the gills are also regulated by pattern recognition receptors (PRRs), including Toll-like receptors (TLRs) among various PRRs. TLRs recognize pathogen-associated molecular patterns (PAMPs), activating downstream signaling pathways that regulate immune responses and the release of antimicrobial substances in crustaceans [8]. Recent studies suggest that terpenoid metabolites generated by FPPS play an immunomodulatory role in combating bacterial and fungal infections. FPPS functions as a regulatory hub, linking the saponin synthesis pathway to immune responses [9]. Consequently, changes in *FPPS* expression within specific tissues like gills may affect localized immune responses, which are crucial for crustaceans to adapt to varying environmental pressures.

*N. denticulata sinensis* is popular in the aquarium market for its vibrant colors, compact size, and easy feeding [10]. Furthermore, it has the characteristics of rapid growth, strong adaptability, and a short life cycle, making it the preferred model for ecological, physiological, and toxicological research [11,12]. In this study, a phylogenetic tree was constructed to analyze the evolutionary relationships of FPPS in *N. denticulata sinensis* (named NdFPPS) and further evaluate the conservation of FPPS among different species. Additionally, we investigated the physicochemical properties, structure, and motifs of FPPS to reveal their evolutionary characteristics. RNA-Seq investigated the potential regulatory mechanisms of *NdFPPS* on the physiological functions after RNA interference (RNAi) to silence the expression of *NdFPPS*. *Vibrio parahaemolyticus* is the main pathogen of acute hepatopancreatic necrosis disease (AHPND) and poses a significant threat to the aquaculture industry [13,14]. Semi-quantitative experiments further evaluated the expression pattern of *NdFPPS* during the early stages of *V. parahaemolyticus* infection (within 24 h), uncovering its potential role in immune regulation within *N. denticulata sinensis*.

## 2. Results

### 2.1. Phylogenetic Analysis

The phylogenetic analysis of FPPS (Figure 1A) reveals that the FPPS of crustaceans, such as *N. denticulata sinensis*, *Penaeus chinensis*, and *Portunus trituberculatus*, exhibited a high degree of conservation, clustering within the same main branch. The internal clustering pattern of crustacean FPPS was consistent with the species phylogenetic tree of crustaceans. The FPPS of *Bombyx mori* clustered closely with that of *Caenorhabditis elegans*, whereas it was more distantly related to the FPPS of *Drosophila melanogaster*, another insect species. In contrast, the FPPS of the vertebrate *Danio rerio* formed a distinct, separate branch. The phylogenetic analysis results indicated that the clustering of FPPS proteins in crustaceans was fully consistent with the phylogenetic clustering results of crustaceans. (Figure 1A and Figure 2). Using the MEME tool, 10 conserved motifs were identified, with motif 2 and motif 3 found in nearly all FPPS. In crustaceans, motif 6 and motif 10 were unique to PvFPPS, motif 8 was exclusive to PclFPPS, and neither motif 5 nor motif 9 was detected in any crustacean FPPS (Figure 1B). All FPPS proteins featured the conserved polyprenyl_synt superfamily domain (Figure 1C). Tandem duplications of *FPPS* were observed in *B. mori* and *Eriocheir sinensis*. Analysis of the *FPPS* structure across different species revealed variations in exon–intron organization. *EsFPPS-2* possessed 10 CDSs, while the *PvFPPS* had only 2 CDSs (Figure 1D).

### 2.2. Physicochemical Properties and Subcellular Localization

The molecular weight (MW) of the FPPS proteins ranged from 37,021.82 Da (PvFPPS) to 53,573.94 Da (MrFPPS), with protein lengths spanning from 327 to 472 amino acids. The theoretical isoelectric point (pI) varied from 5.54 (DrFPPS) to 9.51 (PclFPPS), and the instability index ranged from 38.25 (CeFPPS) to 53.85 (DrFPPS). All FPPS proteins exhibited negative GRAVY values, indicating hydrophilicity. Subcellular localization prediction showed that most FPPS proteins were located in the mitochondria, but there were a few proteins located in the cytoplasm such as CeFPPS, CqFPPS, and DrFPPS (Table 1).

### 2.3. Quality Control and Data Processing

We used the FastQC tool to perform quality control on the raw RNA sequencing data of the gills after knockdown with *NdFPPS*. The FastQC report provided key statistics, including the total sequence count, mean sequence length, median sequence length, percentage of duplicated sequences, and GC content, all of which fell within acceptable ranges (Appendix A). After cleaning and quality filtering, the Q20 and Q30 values exceeded 93%, and the GC content ranged from 37% to 41% (Appendix A), indicating high-quality transcriptome sequencing data.

### 2.4. Identification of Differentially Expressed Genes (DEGs) and Enrichment Analysis

Using DESeq2, 448 DEGs were identified, with 154 upregulated and 294 downregulated genes (Figure 3). The top 10 significantly enriched GO terms were predominantly associated with downregulated genes, suggesting that these biological functions might have been suppressed following *NdFPPS* knockdown (Figure 4A). The majority of the terms in the biological process (BP) category were associated with the regulation of mRNA splicing, pole plasm assembly, and iron ion transport. In the cellular component (CC) category, the majority of the terms were associated with the extracellular matrix, myelin sheath, and clathrin-coated vesicle membrane. In the molecular function (MF) category, the majority of the terms were related to endopeptidase activity, serine hydrolase activity, and serine-type endopeptidase activity (Figure 4B). KEGG pathway analysis revealed the significant enrichment of several immune-related pathways, including the phagosome, MAPK signaling, Toll and Imd signaling, and complement and coagulation cascades. In addition, metabolic pathways such as cholesterol metabolism, amino acid metabolism, and nitrogen metabolism were also enriched (Figure 4C). GSEA indicated negative enrichment scores for multiple immune signaling pathways (Figure 5). COG analysis showed that DEGs were primarily enriched in categories related to signal transduction mechanisms, transcription, translation, ribosomal biogenesis, inorganic ion transport, and metabolism (Figure 6).

### 2.5. Validation of Transcriptome Data by qRT-PCR

DEGs were randomly selected from the transcriptome data for qRT-PCR analysis. The expression trends of these genes were found to be completely consistent with the results obtained from high-throughput sequencing, thereby corroborating the reliability of the transcriptome data (Figure 7).

### 2.6. Expression Profile of NdFPPS Following 24 h Exposure to V. parahaemolyticus

Following *V. parahaemolyticus* infection, the expression of *NdFPPS* displayed a pattern of initial downregulation followed by recovery. During the early infection phase (0–6 h), the *NdFPPS* expression significantly decreased and then gradually returned to normal levels over 6–24 h (Figure 8).

## 3. Discussion

By constructing the phylogenetic tree of FPPS, this study clarified the evolutionary relationships of the FPPS in *N. denticulata sinensis* and other crustaceans (Figure 1A). FPPS from various crustaceans, such as *N. denticulata sinensis*, *P. chinensis*, and *P. trituberculatus*, clustered together on the same main branch, distinctly separated from other species. This separation may be due to the significant role of FPPS in the synthesis of MF, a hormone unique to crustaceans. The clustering of FPPS proteins within crustaceans was highly consistent with the phylogenetic relationships of these species (Figure 2), suggesting that convergent selective pressures may have shaped the evolution of FPPS in these species. This indicates that in crustaceans, species occupying similar ecological niches have experienced comparable environmental selection, and FPPS may play an important role in adapting to these pressures, such as regulating immunity metabolism and hormone synthesis, ultimately aiding in survival and reproduction within specific ecological environments. Furthermore, despite *B. mori* and *C. elegans* belonging to distinct phyla (Arthropoda and Nematoda, respectively) [15,16], their FPPS clustered closely, indicating a conserved function across these lineages. In contrast, the FPPS of the vertebrate *D. rerio* formed an independent branch, reflecting the evolutionary divergence in the FPPS pathway between vertebrates and invertebrates.

By analyzing the structure, conserved domains, and motifs of FPPS, this study revealed the evolutionary characteristics of FPPS across species. There were significant differences in the exon–intron structures of *FPPS*, especially in crustaceans. For example, *EsFPPS-2* contained 10 CDSs, while *PvFPPS* only had two. Using the MEME motif search tool, 10 conserved motifs were identified and their distribution across FPPS was analyzed. The results indicated that these motifs ranged from 17 to 143 amino acids, with motif 1 containing the fewest amino acids and motif 10 containing the most. Motif 2 and motif 3 were present in nearly all FPPS. Additionally, the conserved motif distribution pattern in crustacean FPPS was similar yet showed significant divergence from other species, reflecting the unique role of crustacean FPPS in MF synthesis. During evolution, FPPS in Penaeus gained motif 7. Notably, *P. vannamei* lost motifs 1 and 3 while gaining motifs 6 and 10, which were absent in other crustacean FPPS. Furthermore, FPPS in *B. mori* and *E. sinensis* exhibited tandem duplication, suggesting potentially higher metabolic demands in these species.

GO enrichment analysis indicated that immune-related functions, such as iron ion metabolism, protein degradation, and lipoprotein binding, were significantly suppressed following the knockdown of the expression of *NdFPPS*. This suggests that FPPS may play a regulatory role in maintaining immune balance and cellular metabolism. For example, iron ion metabolism plays a crucial role in crustacean immunity by limiting iron availability to pathogens, inducing oxidative stress, and regulating immune signaling pathways to enhance resistance against infections [17].

KEGG enrichment analysis revealed the significant enrichment of multiple immune-related pathways, including the phagosome, MAPK signaling, Toll and Imd receptor signaling, and complement and coagulation cascade pathways. The phagosome pathway plays a critical role in the immune system, primarily responsible for recognizing, engulfing, and degrading foreign pathogens. Through this process, it initiates pathogen clearance at the cellular level, thereby contributing to the maintenance of host immune homeostasis [18,19]. In crustaceans, the MAPK signaling pathway plays a central role in immune defense by responding to pathogen-associated stimuli to regulate genes involved in cell proliferation, differentiation, and stress responses, thereby enhancing immune function [20,21]. The Toll and Imd pathways recognize PAMPs, triggering the release of antimicrobial peptides and activating inflammatory responses to defend against pathogen invasion [22]. The significant enrichment of the complement and coagulation cascade pathway suggests that FPPS may be involved in the regulation of responses to pathogen infection or cellular damage. However, the full functionality of the complement cascade in crustaceans remains uncertain and requires further investigation [23]. In addition, some downregulated genes were enriched in the immune-related NOD-like receptor (NLR) and Ras signaling pathways. NLRs, as intracellular pathogen-recognition receptors, activate inflammation and the release of antimicrobial peptides and immune molecules upon recognizing PAMPs, enhancing host resistance to infection [24]. The Ras signaling pathway modulates various immune processes, including cell proliferation, adhesion, and phagocytosis, aiding crustaceans in recognizing and responding to foreign pathogens. In particular, under pathogenic bacterial stimulation, the Ras pathway can regulate the expression of genes related to inflammation, antimicrobial peptide production, and reactive oxygen species generation [25]. GSEA enrichment analysis showed negative enrichment scores for pathways such as the complement and coagulation cascades, MAPK signaling, NOD-like receptor signaling, Toll and Imd signaling, and Ras signaling, indicating that these immune-related pathways were suppressed following the knockdown of the expression of *NdFPPS*.

Additionally, the significant enrichment of metabolic pathways, particularly cholesterol metabolism, amino acid metabolism, and nitrogen metabolism, suggests potential impacts on cellular membrane composition, immune molecule synthesis, and energy supply, thereby influencing immune cell activity and overall immune response. Alterations in cholesterol metabolism may affect membrane fluidity and structural integrity, thereby impacting cell signaling and immune responses [26]. The enrichment of amino acid metabolism suggests that amino acid synthesis or degradation may be affected, subsequently influencing immune cell proliferation and activation, thereby impacting the organism’s ability to defend against pathogenic interference [27]. Changes in the nitrogen metabolism pathway may affect the supply of amino acids. These enrichment results indicate that the knockdown of the expression of *NdFPPS* may impact fundamental metabolic processes and could indirectly affect the immune response mechanisms in *N. denticulata sinensis*.

COG functional classification analysis revealed the significant enrichment of DEGs in categories such as signal transduction, gene transcription regulation, protein synthesis, and inorganic ion transport and metabolism. These enrichment results suggest that the knockdown of the expression of *NdFPPS* may impact various physiological functions in gills, including immune regulation, gene expression, and environmental response mechanisms.

*V. parahaemolyticus* infection in shrimp typically progresses through four key stages of pathogenesis. Initially, the bacteria enter the host through water or feed, colonizing the gill and oral mucosal regions and then rapidly spreading to the bloodstream and other critical organs, thereby activating the host immune system [28]. In the subsequent acute phase, rapid bacterial proliferation leads to hepatopancreatic necrosis, causing visible symptoms such as reduced appetite and paler body coloration. In the terminal phase, organ failure and immune system collapse result in mass mortality, especially in high-density farming environments [29,30]. In this study, *NdFPPS* was significantly downregulated within 6 h after *V. parahaemolyticus* challenge, indicating that *NdFPPS* may play an important role in the initial immune response. During the early stages of infection, the immune system may quickly mobilize various defense mechanisms to combat pathogen invasion. During this time, cells may prioritize certain stress responses to manage the infection, thereby temporarily suppressing *NdFPPS* expression. It is speculated that within 6 to 24 h of the challenge, the bacteria may spread to other organs, fully activating the immune system of the host to combat the pathogen. During this phase, *NdFPPS* expression gradually returns to its original levels, suggesting that the host reallocates metabolic resources to support an active immune state. This expression trend further indicates that the metabolic pathways involving *NdFPPS* may indirectly regulate immune responses. This conclusion provides essential insights for future research into the relationship between FPPS, immune regulation, and metabolic homeostasis in crustaceans.

In summary, this study analyzed the evolutionary characteristics of the *NdFPPS* across species, including *N. denticulata sinensis*, by constructing a phylogenetic tree. Transcriptomic analysis of gills indicated that knockdown of the expression of *NdFPPS* might impact fundamental metabolic processes and indirectly influence immune response capacity. Under *V. parahaemolyticus* stimulation, the expression of *NdFPPS* was significantly downregulated during the early stage (0–6 h), followed by a gradual recovery to normal levels between 6 and 24 h. These results further suggest that *NdFPPS* may play an important role in immune response regulation in *N. denticulata sinensis*, but further functional validation is required to confirm this conclusion.

## 4. Materials and Methods

### 4.1. Animals

Healthy shrimp (*N. denticulata sinensis*) with 2.0 ± 0.5 cm body lengths were purchased from a local market in Anxin County, Baoding, Hebei Province, China. The shrimp were cultured at 25 ± 1 °C (light:dark = 1:1) in tanks with a recirculation system for seven days before experimentation. The shrimp were fed with a commercial diet (Sera shrimps natural, Kunshan) twice a day.

### 4.2. Phylogenetic Analysis, Physicochemical Properties, and Subcellular Localization Prediction

The hidden markov model (HMM) matrix file for the FPPS domain polyprenyl_synt (PF00348) was obtained from the Pfam database (http://pfam.xfam.org/, accessed on 11 September 2024) and subsequently searched against the genome of *N. denticulata sinensis* using HMMER (v3.0) software (genome data were generated in our laboratory and are currently unpublished). Candidate gene sequences were screened and further validated through BLASTP on the NCBI website (https://www.ncbi.nlm.nih.gov/, accessed on 13 September 2024). Based on the results from HMMER and BLASTP, candidate protein sequences were confirmed for the FPPS domain using the Pfam (http://pfam.xfam.org/, accessed on 11 September 2024) and CDD databases (https://www.ncbi.nlm.nih.gov/Structure/bwrpsb/bwrpsb.cgi, accessed on 13 September 2024), ultimately identifying the FPPS sequence for *N. denticulata sinensis*. For phylogenetic analysis, FPPS sequences from *D. melanogaster*, *D. rerio*, *B. mori*, and *C. elegans* were retrieved from the NCBI database as outgroups and analyzed along with FPPS sequences from various crustaceans to explore evolutionary relationships. Multiple sequence alignments were performed on the full-length protein sequences using MUSCLE (v5.1) [31]. Phylogenetic trees were constructed by maximum likelihood (ML) analysis with 1000 bootstrap replicates using IQ-TREE (v2.2.2.2) software [32]. To illustrate the phylogenetic relationships among crustaceans, gene families were identified using OrthoFinder (v2.5.4) with DIAMOND as the alignment tool. Single-copy genes identified by OrthoFinder (v2.5.4) were aligned end-to-end using MAFFT (v7.427). Subsequently, a maximum-likelihood phylogenetic tree was constructed with RAxML (v8.2.12) using the PROTGAMMAJTT model and performing 1000 bootstrap replicates. Finally, divergence times were estimated using MCMCTREE (v4.9) [33]. Additionally, the molecular weight, instability index, and theoretical pI of FPPS proteins from each species were computed using TBtools-II software [34], while subcellular localization predictions were performed with the online tool WoLF PSORT (https://wolfpsort.hgc.jp/, accessed on 15 September 2024) [35].

### 4.3. Gene Structure and Conserved Motifs Analysis

Conserved motif analysis was conducted using the MEME Suite online tool (https://meme-suite.org/, accessed on 18 September 2024) [36]. The positions of exons and CDSs on chromosomes were extracted through custom scripts. For domain prediction, conserved domains within candidate proteins were identified using the NCBI CD-Search tool (https://www.ncbi.nlm.nih.gov/Structure/cdd/wrpsb.cgi, accessed on 18 September 2024). Finally, to comprehensively illustrate the evolutionary relationships and functional characteristics of the proteins, TBtools-II software was used to generate a four-in-one visualization that integrated data from phylogenetic trees, gene structure, conserved domains, and conserved motif analyses.

### 4.4. Double-Stranded RNA (dsRNA) Synthesis and Injection

Primers were designed based on the ORF sequences of *NdFPPS* and *EGFP*, with T7 promoter sequences appended to the 5′ ends for in vitro transcription (Appendix A). Using the TranscriptAid T7 High Yield Transcription Kit from (Thermo Scientific, Waltham, MA, USA), ds*NdFPPS* and ds*EGFP* were synthesized and purified. According to our previous research [5], ds*NdFPPS* was injected at 0.4 μg/μL using a sterile syringe into the second abdominal segment of the shrimp. Correspondingly, a control group was injected with the same dose of ds*EGFP*. Gills were collected from the experimental group and control group 12 h after the injection of dsRNA. Each sample comprised three biological replicates, designated as E1, E2, E3 for the experimental group and C1, C2, and C3 for the control group. All samples were immediately frozen in liquid nitrogen and stored at −80 °C for subsequent RNA sequencing.

### 4.5. RNA Extraction, Library Construction, Sequencing, and Data Processing

Total RNA was isolated from the gills, including three replicates of the ds*NdFPPS*-injected group (experimental group: E1, E2, E3) and the *dsEGFP*-injected group (control group: C1, C2, C3) 12 h post-injection. The RNA extraction was conducted using TRIzol Reagent (Invitrogen, Carlsbad, CA, USA) and treated with RNase-free DNase I (TAKARA, Dalian, China) to eliminate genomic DNA contamination. The quality and integrity of the RNA were assessed using 1% agarose gels for degradation and contamination, and purity was verified with a NanoPhotometer^®^ spectrophotometer (IMPLEN, Westlake Village, CA, USA). The RNA integrity was further evaluated using the Bioanalyzer 2100 system with the Nano 6000 Assay Kit (Agilent Technologies, Santa Clara, CA, USA). Each sample was prepared with 1 μg of total RNA for the RNA sample preparations. Sequencing libraries were constructed using the NEBNext^®^ Ultra™ RNA Library Prep Kit for Illumina^®^ (NEB, Ipswich, MA, USA). Following the manufacturer’s instructions, index-coded samples were clustered using a cBot Cluster Generation System using the TruSeq PE Cluster Kit v3-cBot-HS (Illumina, San Diego, CA, USA). Post-clustering, the libraries were sequenced on an Illumina Novaseq 6000 system, capturing the fluorescence signals, which were then converted into sequencing reads by computer software. This process generated 150 bp paired-end reads.

Initial quality assessments of all raw sequencing data were performed using FastQC (v0.11.9), and the results were compiled with MultiQC (v1.6) for an overview analysis. Data cleaning was then conducted with the software fastp (v0.23.4), which included the removal of adapter sequences and low-quality reads, yielding clean reads for subsequent analysis. The quality-controlled clean reads were aligned to the *N. denticulata sinensis* reference genome using HISAT2 (v2.2.1) [37]. The reference genome and annotation files were generated by our team. Alignment outputs were generated in SAM format and subsequently converted to BAM format using Samtools (v1.18), followed by sorting and indexing of the BAM files.

### 4.6. Gene Quantification and Functional Enrichment of DEGs

HTSeq was used to quantify gene expression from BAM files, producing a read count matrix for each sample [38]. The matrix was then normalized to fragments per kilobase per million mapped reads (FPKM) for analysis. Sample correlation was evaluated using Spearman’s rank correlation coefficients via the corrplot package in R. DEGs were identified with DESeq2 in R, applying an FDR < 0.05 and |log2(FoldChange)| > 1. Protein sequences from *N. denticulata sinensis* were annotated through eggNOG-mapper based on the eggNOG database (http://eggnogdb.embl.de/#/app/emapper, accessed on 27 August 2024). An OrgDb database specific to *N. denticulata sinensis* was constructed using the AnnotationForge package [39], supporting functional annotation. Enrichment analysis for DEGs was conducted with clusterProfiler in R, encompassing GO, KEGG, and GSEA analyses. GO and KEGG enrichments were tested using the Hypergeometric Distribution Test, while GSEA used KEGG pathways as gene sets. Moreover, downregulated genes underwent Cluster of Orthologous Groups (COG) functional annotation.

### 4.7. Validation Using qRT-PCR

The accuracy of the transcriptome sequencing results was validated through qRT-PCR. Five downregulated and two upregulated genes were randomly selected for qRT-PCR validation. Due to the stable expression of the *18S* rRNA (GenBank accession No. OP185352), it was used as the internal reference gene. Seven pairs of specific primers (Appendix A) were designed based on the transcriptome sequences. The protocol began with an initial denaturation at 95 °C for 3 min, followed by 45 cycles each consisting of 10 s at 95 °C, 10 s at 55/60 °C, and 15 s at 72 °C (Bioer LineGene 9600, Hangzhou, China). Furthermore, to ensure that the amplification product was a single product, melting curves were generated from 60 °C to 95 °C at a rate of 0.4 °C/s. All reactions were performed in triplicate, including the no-template control (NTC). The relative expression was quantified using the comparative CT method. The expression of the target gene in the experimental group was compared to that in the control group using the CT (2^−ΔΔCT^) method [40].

### 4.8. Expression Profile of NdFPPS in Gills After V. parahaemolyticus Challenge

In this study, the *V*. *parahaemolyticus* challenge test used previously described methods [41]. Briefly, *V*. *parahaemolyticus* was inoculated in LB liquid medium and incubated overnight at 37 °C with 200 r/min shaking. The bacteria were collected by centrifugation at 2000 r/min for 5 min and resuspended in 1× PBS buffer (Sangon, Beijing, China) to obtain *V*. *parahaemolyticus* at a concentration of 1 × 10^6^ CFU/mL. We divided 60 shrimps evenly into three groups and injected 5 μL of live *V*. *parahemolyticus* at a concentration of 1 × 10^6^ CFU/mL into each shrimp in the group. Three individuals from each group were randomly sampled at 0, 6, 12, and 24 h post-challenge. After that, the gills were collected for total RNA extraction.

### 4.9. Statistical Analysis

All data were expressed as the mean ± standard deviation (*n* = 3), and statistical analysis was performed using SPSS Statistics 19. A one-way analysis of variance (ANOVA) and Duncan’s multiple tests were used for data analysis in gene expression and relative activity. We used a *t*-test to compare the statistical significance between the ds*EGFP* and ds*NdFPPS* treatment groups. The data for this study were processed using GraphPad Prism software (Version 9.0.2, San Diego, CA, USA).

## Figures and Tables

**Figure 1 ijms-26-00065-f001:**
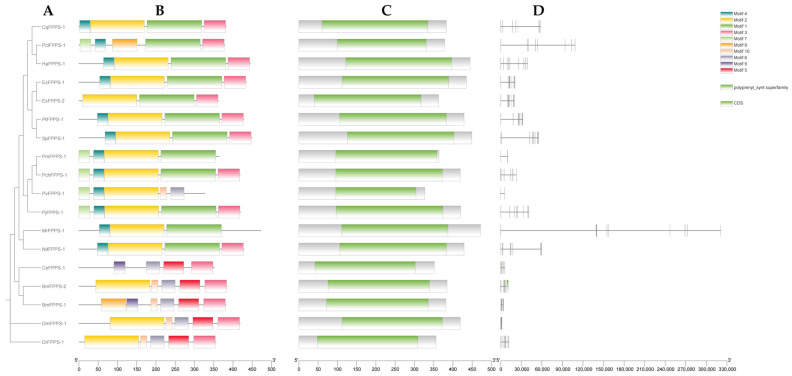
The structure and motif analysis of FPPS. (**A**) The phylogenetic relationships of FPPS across different species, represented as a dendrogram. The FPPS proteins of different species include *B.mori* (BmFPPS-1, BmFPPS-2), *C. elegans* (CeFPPS), *Cherax quadricarinatus* (CqFPPS), *D. melanogaster* (DmFPPS), *D. rerio* (DrFPPS), *E. sinensis* (EsFPPS-1, EsFPPS-2), *Homarus americanus* (HaFPPS), *Macrobrachium rosenbergii* (MrFPPS), *P. chinensis* (PchFPPS), *P. japonicus* (PjFPPS), *P. monodon* (PmFPPS), *P. trituberculatus* (PtFPPS), *P. vannamei* (PvFPPS), *Procambarus clarkii* (PclFPPS), and *Scylla paramamosain* (SpFPPS). (**B**) The distribution of conserved motifs within FPPS proteins, with different colors representing distinct motifs (motif 1 to motif 10). (**C**) A representation of the polyprenyl_synt superfamily domain, with green indicating conserved catalytic domains across species. (**D**) The structure of FPPS genes, with green boxes representing coding sequences (CDS). The GenBank accession numbers for the sequences mentioned in the phylogenetic tree are detailed in Appendix A.

**Figure 2 ijms-26-00065-f002:**
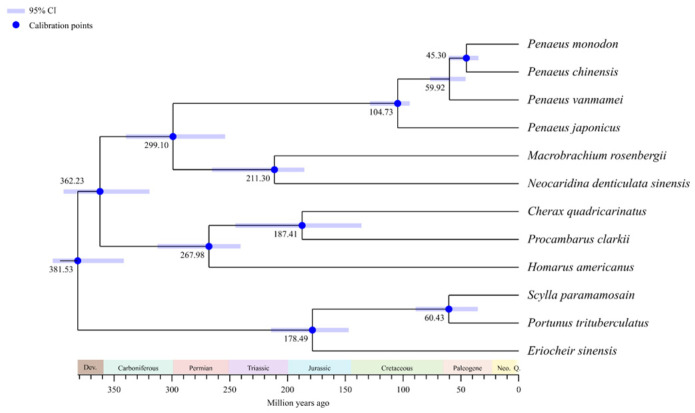
A phylogenetic time tree of crustacean species. This time tree illustrates the evolutionary divergence times among various crustacean species. Blue circles indicate calibration points, and horizontal blue lines represent 95% confidence intervals. The timeline, marked in millions of years ago (Mya), spans the evolutionary divergence from the Carboniferous period to the Neogene of the Cenozoic era. The GenBank accession numbers for the genome data mentioned in the phylogenetic tree are detailed in Appendix A.

**Figure 3 ijms-26-00065-f003:**
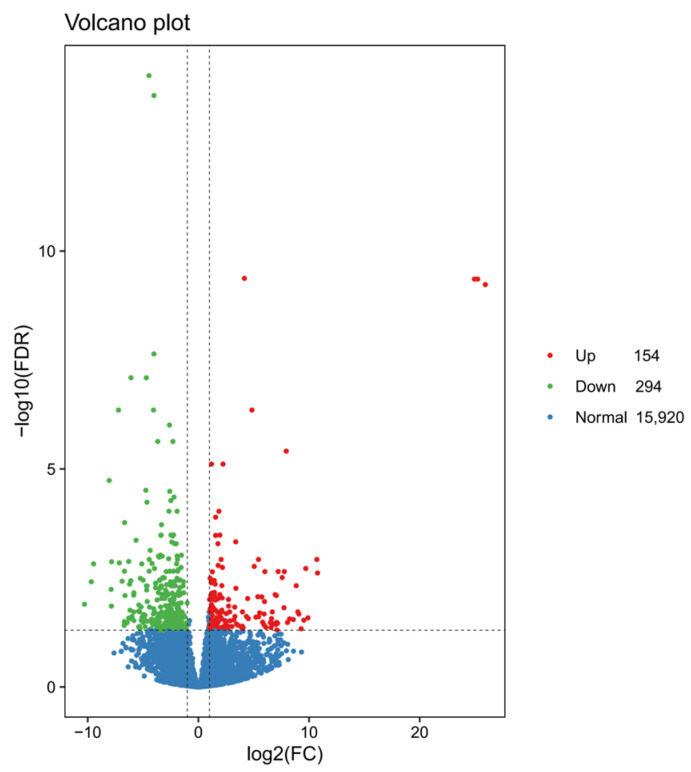
A volcano plot showing the distribution of DEGs. This volcano plot illustrates the distribution of gene expression changes. The *x*-axis represents the fold change in gene expression (log2(FC)), and the *y*-axis shows the significance level (−log10(FDR)). Red dots indicate significantly upregulated genes (Up), green dots represent significantly downregulated genes (Down), and blue dots denote genes without significant differential expression (Normal).

**Figure 4 ijms-26-00065-f004:**
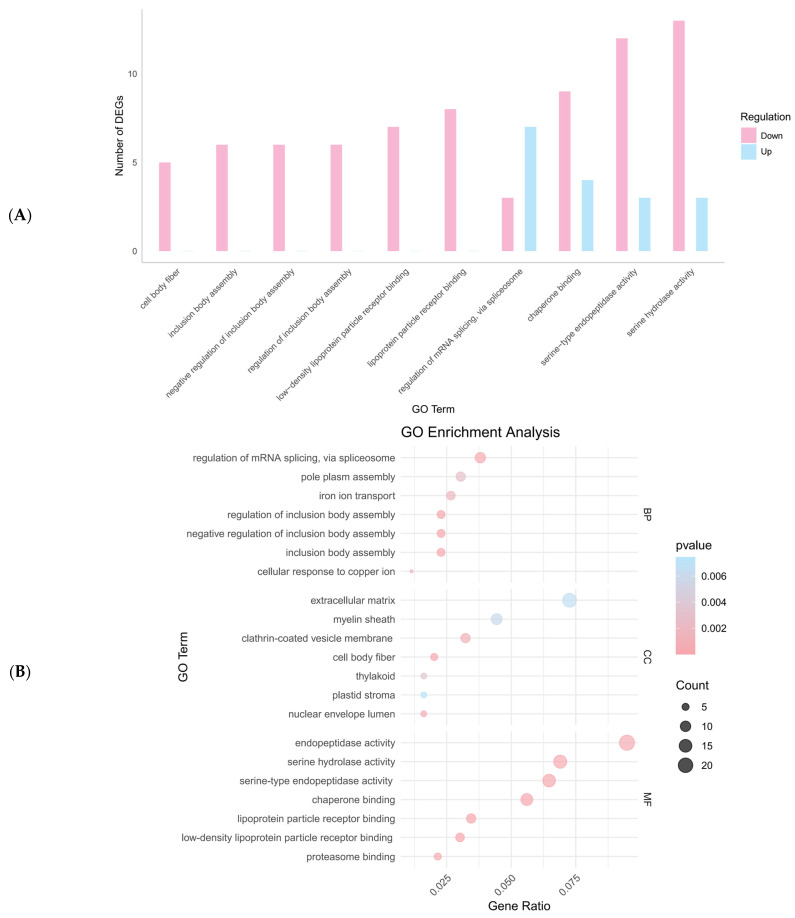
Functional and pathway enrichment analysis of DEGs. (**A**) The GO classification shows the top 10 enriched GO terms for upregulated and downregulated DEGs across the BP, CC, and MF categories. Blue bars represent upregulated genes, while pink bars represent downregulated genes. (**B**) GO enrichment analysis of DEGs, with each dot size indicating the number of genes and the color intensity representing the *p*-value (darker colors correspond to smaller *p*-values). The gene ratio represents the ratio of DEGs in each GO term to the total DEGs analyzed. (**C**) KEGG pathway enrichment analysis for all DEGs, displaying significantly enriched pathways. The dot size indicates the gene count, while the color intensity indicates the *p*-value significance.

**Figure 5 ijms-26-00065-f005:**
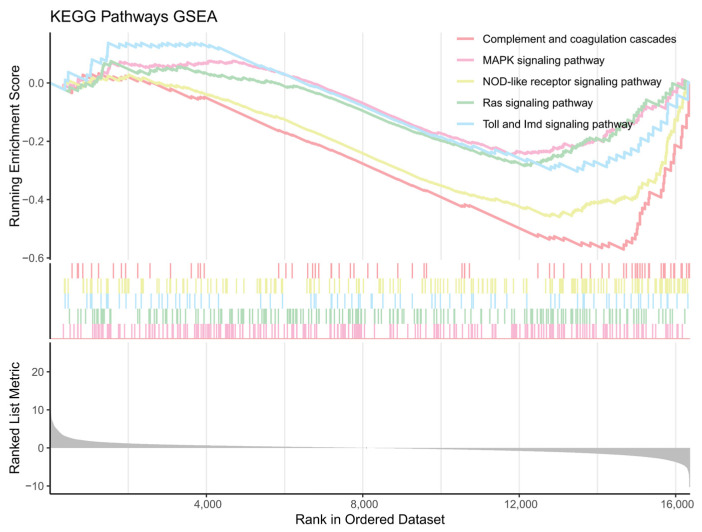
GSEA enrichment analysis of KEGG pathways. This figure presents the GSEA enrichment curves for five immune-related KEGG pathways. The top curves illustrate the changes in enrichment scores for each pathway across gene rankings, with the *x*-axis representing the ranked gene positions and the *y*-axis showing the running enrichment scores. The middle bar chart displays the gene distribution for each pathway within the ranked dataset.

**Figure 6 ijms-26-00065-f006:**
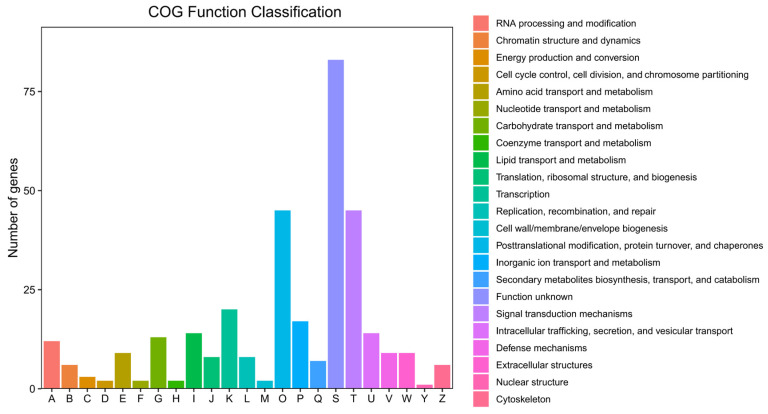
COG functional classification. This figure displays the distribution of downregulated DEGs across various COG functional categories. The *x*-axis shows COG category labels, while the *y*-axis indicates the gene count in each category. Different colors represent distinct COG functions.

**Figure 7 ijms-26-00065-f007:**
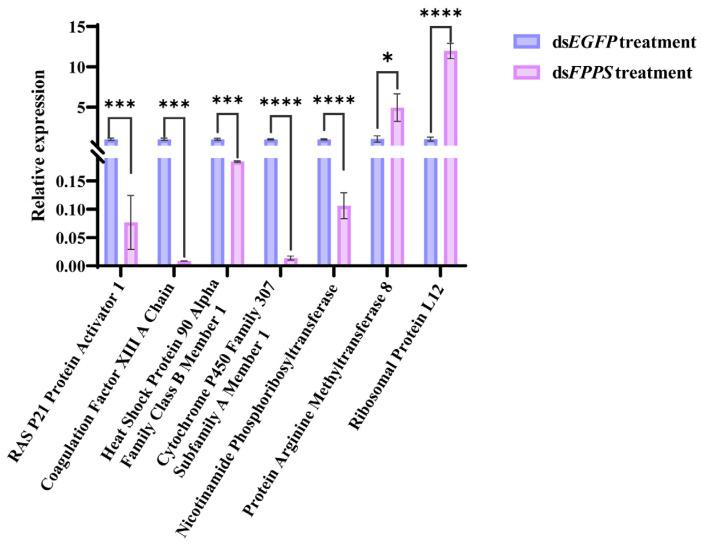
Relative expression of seven DEGs. Asterisks indicate significant differences in seven selected genes’ expression compared to control group, respectively (*: *p* < 0.05; ***: *p* < 0.001; and ****: *p* < 0.0001).

**Figure 8 ijms-26-00065-f008:**
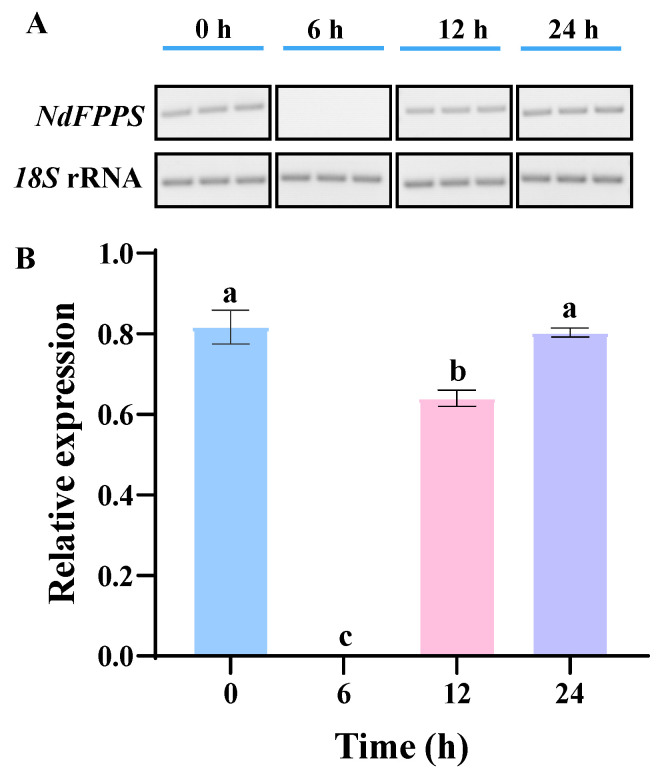
The semi-quantitative validation of the *NdFPPS* expression pattern in the early stage of *V. parahaemolyticus* challenge. (**A**) Gel maps of *NdFPPS* and *18S* rRNA genes challenged by *V. parahaemolyticus* at different times. (**B**) Significance analysis of grayscale values in gel results. The transcript expression levels of *NdFPPS* were normalized to the *18S* rRNA gene. The different letters on the column represent statistical significance (*p* < 0.05), and the same letters indicate no significance among groups (*p* > 0.05).

**Table 1 ijms-26-00065-t001:** Biophysical properties and subcellular localization of FPPS proteins from different species.

Protein	Amino Acid Length	MW	pI	Instability Index	Aliphatic Index	GRAVY	Subcellular Location
BmFPPS-1	382	44,788.89	8.77	42.79	84.21	−0.368	Mitochondria
BmFPPS-2	385	44,086.73	7.96	39.02	88.44	−0.268	Mitochondria
CeFPPS	352	40,266.39	8.09	38.25	90.00	−0.274	Cytoplasm
CqFPPS	383	43,348.01	8.56	47.04	91.17	−0.176	Cytoplasm
DmFPPS	419	47,903.13	6.47	41.35	89.36	−0.238	Mitochondria
DrFPPS	356	40,823.73	5.54	53.85	87.7	−0.305	Cytoplasm
EsFPPS-1	435	48,927.19	9.31	48.5	83.45	−0.257	Mitochondria
EsFPPS-2	363	41,350.85	8.66	43.22	86.28	−0.252	Cytoplasm
HaFPPS	445	50,234.68	8.72	44.44	90.74	−0.214	Mitochondria
MrFPPS	472	53,573.94	8.87	51.31	86.57	−0.199	Mitochondria
PchFPPS	419	47,232.14	8.39	50.76	87.57	−0.224	Mitochondria
PclFPPS	379	42,660.19	9.51	50.79	84.62	−0.218	Mitochondria
PjFPPS	420	47,334.21	8.79	49.51	87.57	−0.263	Mitochondria
PmFPPS	364	40,979.87	6.87	48.84	85.27	−0.247	Mitochondria
PtFPPS	429	47,808.05	8.97	46.24	88.51	−0.162	Mitochondria
PvFPPS	327	37,021.82	8.60	49.56	94.89	−0.073	Mitochondria
SpFPPS	449	50,150.75	9.23	46.53	88.02	−0.190	Mitochondria
NdFPPS	429	48,250.54	8.88	48.63	84.80	−0.234	Mitochondria

Note: The FPPS proteins of the studied species include *B.mori* (BmFPPS-1, BmFPPS-2), *C. elegans* (CeFPPS), *Cherax quadricarinatus* (CqFPPS), *D. melanogaster* (DmFPPS), *D. rerio* (DrFPPS), *E. sinensis* (EsFPPS-1, EsFPPS-2), *Homarus americanus* (HaFPPS), *Macrobrachium rosenbergii* (MrFPPS), *P. chinensis* (PchFPPS), *P. japonicus* (PjFPPS), *P. monodon* (PmFPPS), *P. trituberculatus* (PtFPPS), *P. vannamei* (PvFPPS), *Procambarus clarkii* (PclFPPS), and *Scylla paramamosain* (SpFPPS).

## Data Availability

The raw sequence data reported in this paper have been deposited in the Genome Sequence Archive in the National Genomics Data Center, China National Center for Bioinformation/Beijing Institute of Genomics, Chinese Academy of Sciences (GSA: CRA018357) (https://ngdc.cncb.ac.cn/?lang=zh, accessed on 26 September 2024).

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
