# Peer review of "Transcriptomic Analysis of Neocaridina denticulata sinensis Gills Following FPPS Knockdown Reveals Its Regulatory Role in Immune Response"

_ijms, 2024, doi:10.3390/ijms26010065_

Round 1

Reviewer 1 Report

Comments and Suggestions for Authors

In this study, authors analyzed the evolutionary differences in the physicochemical properties, subcellular localization, gene structure, and motif composition of NdFPPS in Neocaridina denticulata sinensis compared to other species. Significant evolutionary divergence of FPPS was observed in crustaceans, likely linked to its role in MF synthesis. After RNA interference (RNAi)-mediated knockdown of NdFPPS, transcriptomic analysis of gills revealed significant enrichment of differentially expressed genes (DEGs) in pathways related to metabolism and immunity. The findings are significant, the experimental methods and results are acceptable. However, to improve the quality of the manuscript, there are some problems should be addressed before it is accepted, which are listed below.

Question 1: section 2.2, Mito and Cyto are not mentioned again in this manuscript. It is suggested to delete the abbreviations.

Question 2: The manuscript in Figure 2 is vague, it is recommended to replace it with a high-definition figure.

Question 3: In the volcano plot of Figure 3, it is better to clearly label the number of up-regulated, down-regulated, and without significant differential expression.

Question 4: The manuscript lacks a description of MF categories. It is suggested to add a description of differentially expressed genes in MF categories to make the article more complete.

Question 5: The font of “′” should be consistent with the entire text.

Question 6: Table 1 involves the characteristics of proteins. It is recommended to change the genes in Table 1 to proteins.

Question 7: It is recommended to insert the 38th reference after the sentence HTSeq was used to quantify gene expression from BAM files, producing a read count matrix for each sample.

Author Response

Comments 1: [Section 2.2, Mito and Cyto are not mentioned again in this manuscript. It is suggested to delete the abbreviations.]

Response 1: [Mito and Cyto have been removed from the revised manuscript] Thank you for pointing this out. We agree with this comment. Therefore, [Mito and Cyto have been removed from page 4 of the revised manuscript.]

Comments 2: [The manuscript in Figure 2 is vague, it is recommended to replace it with a high-definition figure.]

Response 2: [We have replaced the high-resolution Figure 2 according to the reviewer's suggestion.] Thank you for pointing this out. We agree with this comment. Therefore, [We have replaced the high-definition Figure 2, which can now be found on page 3 of the revised manuscript.]

Comments 3: [In the volcano plot of Figure 3, it is better to clearly label the number of up-regulated, down-regulated, and without significant differential expression.]

Response 3: [We have added the numbers of upregulated, downregulated, and normally expressed genes to the volcano plot.] Thank you for pointing this out. We agree with this comment. Therefore, [Please refer to the revised manuscript on page 5 for details.]

Comments 4: [The manuscript lacks a description of MF categories. It is suggested to add a description of differentially expressed genes in MF categories to make the article more complete.]

Response 4: [As requested by the reviewer, the GO annotations of DEGs have been elaborated in detail.] Thank you for pointing this out. We agree with this comment. Therefore, [The MF categories have been elaborated in detail, please refer to page 4 of the revised manuscript for details.]

Comments 5: [The font of “′” should be consistent with the entire text.]

Response 5: [The font of “′” has been adjusted and is consistent with the font of the entire text.] Thank you for pointing this out. We agree with this comment. Therefore, [The font of “′” has been changed to the Palatino Linotype font, which is consistent with the font of the entire text. Please refer to page 12 of the revised manuscript for details.]

Comments 6: [Table 1 involves the characteristics of proteins. It is recommended to change the genes in Table 1 to proteins.]

Response 6: [According to the reviewer's request, the gene in Table 1 has been changed to protein.] Thank you for pointing this out. We agree with this comment. Therefore, [The first column of the gene in Table 1 has been changed to protein, as detailed in Table 1 of the revised manuscript.]

Comments 7: [It is recommended to insert the 38th reference after the sentence “HTSeq was used to quantify gene expression from BAM files, producing a read count matrix for each sample”.]

Response 7: [The 38th reference has been inserted after “HTSeq was used to quantify gene expression from BAM files, producing a read count matrix for each sample”.] Thank you for pointing this out. We agree with this comment. Therefore, [The 38th reference has been inserted after the sentence “HTSeq was used to quantify gene expression from BAM files, producing a read count matrix for each sample”. For details, please refer to page 12 of the revised manuscript.]

Reviewer 2 Report

Comments and Suggestions for Authors

Manuscript: ijms-3328490

1.     This is a straightforward and well-written manuscript. The authors conducted transcriptomic analysis of Neocaridina denticulata sinensis gills following FPPS knockdown.

2.     The authors presented structure and motif of FPPS (Fig.1), phylogenetic time tree (Fig. 2), biophysical properties (Table 1), volcano plot (Fig. 3), functional and pathway enrichment analysis (Fig. 4), GSEA enrichment analysis (Fig. 5), COG functional classification (Fig. 6), relative expression (Fig. 7), semi-quantitative validation FdFPPS epression (Fig. 8).

3.     The authors concluded and reported the expression of NdFPPS was significantly downregulated during the early time (0-6 h) and reported that NdFPPS may play an important role in immune response regulation in N. denticulata sinensis.

Author Response

Comments 1: [The reviewer did not provide any comments on this research article.]

Response 1: [Not applicable.]

Reviewer 3 Report

Comments and Suggestions for Authors

The manuscript submitted by Dr. Li et al. described the evolutionary characteristics of NdFPPS. The authors further investigated the regulatory mechanism of NdFPPS using RNAi and RNAseq techniques. This study is scientific sounds, and the experiment is well-designed. Some minor concerns are listed below:

1. Abstract: “…. and motif composition of FPPS (named NdFPPS) in Neocaridina denticulata sinensis compared to other species” is recommended changing to “…. and motif composition of FPPS in Neocaridina denticulata sinensis (named NdFPPS) compared to other species.”

Same suggestion for the last paragraph in the Introduction section.

2. Table 1: 1st and 2nd column names should be “Protein” and “Amino Acid Length.”

3. Figure 1 and Table 1: The abbreviations for each species of FPPS are recommended to explain in the legends.

4. 2.3. section: No supplementary Material Tables are found.

5. “4.9. Statistical Analysis” is needed.

Author Response

Comments 1: [Abstract: “…. and motif composition of FPPS (named NdFPPS) in Neocaridina denticulata sinensis compared to other species” is recommended changing to “…. and motif composition of FPPS in Neocaridina denticulata sinensis (named NdFPPS) compared to other species.”.]

Response 1: [The abstract and introduction sections have been revised according to the suggested modification.] Thank you for pointing this out. We agree with this comment. Therefore, [For specific modification details, please refer to page 1 and page 2.]

Comments 2: [Table 1: 1st and 2nd column names should be “Protein” and “Amino Acid Length.”]

Response 2: [The names of 1st and 2nd columns in Table 1 have been revised to "Protein" and "Amino Acid Length"] Thank you for pointing this out. We agree with this comment. Therefore, [The names of the first and second columns in Table 1 have been changed to "Protein" and "Amino Acid Length", as detailed in Table 1 of the revised manuscript.]

Comments 3: [Figure 1 and Table 1: The abbreviations for each species of FPPS are recommended to explain in the legends.]

Response 3: [The abbreviations for each species FPPS have been explained respectively in the legends of Table 1 and Figure 2.] Thank you for pointing this out. We agree with this comment. Therefore, [For specific revisions, please refer to page 3 and page 4 of the revised manuscript.]

Comments 4: [4. 2.3. section: No supplementary Material Tables are found.]

Response 4: [We have successfully uploaded the supplementary materials and speculate that supplementary materials may not have been discovered at the corresponding places.] Thank you for pointing this out. We agree with this comment. Therefore, [The supplementary materials have been successfully uploaded.]

Comments 5: [“4.9. Statistical Analysis” is needed.]

Response 5: [We have added a section titled "4.9. Statistical Analysis" to the “4. Materials and methods” section of the revised manuscript.] Thank you for pointing this out. We agree with this comment. Therefore, [The specific statement has been added to the revised manuscript, please refer to page 13 for details.]
